# Responsible Reasoning with Large Language Models and the Impact of Proper Nouns

**Sumit Kumar Jha**
Computer Science Department
University of Texas at San Antonio
San Antonio, TX, 78255, USA
sumit.jha@utsa.edu

**Alvaro Velasquez**
Department of Computer Sience
University of Colorado Boulder
Boulder, CO, 80309, USA
alvaro.velasquez@colorado.edu

**Rickard Ewetz**
ECE Department
University of Central Florida
Orlando, FL, 32816, USA
ewetz@eecs.ucf.edu

**Susmit Jha**
Computer Science Laboratory
SRI International
Menlo Park, CA, 94025, USA
susmit.jha@sri.com

## Abstract

Language models with billions of parameters have shown remarkable emergent properties, including the ability to reason on unstructured data. We show that open-science multi-lingual large language models can perform the task of spatial reasoning on two or more entities with significant accuracy. A responsible large language model would perform this spatial reasoning task with the same accuracy regardless of the choice of the names of the entities over which the spatial relationships are defined. However, we show that the accuracies of contemporary large language models are impacted by the choice of proper nouns even when the underlying task ought to be independent of the choice of proper nouns. In this context, we observe that the conditional log probabilities or beam scores of open-science multi-lingual large language model predictions are not well-calibrated, and the beam scores do not discriminate well between correct and wrong responses in this context.

## 1 Introduction

Over the last four decades, cognitive psychologists and neurolinguistic studies have investigated the impact of proper names on human cognition with interesting outcomes (Bredart et al., 2002). It is now widely accepted that human beings find it more difficult to recall personal names (Young et al., 1985, 1988) than other kinds of words, including relatively rare common nouns. Several different explanations have been investigated for such a discrepancy, including (i) the semantic tag nature of personal names without being descriptive (Fogler & James, 2007), (ii) the need to obtain a single correct label with no synonyms (Hanley, 2011), (iii) the larger set of possible phonologies (James & Fogler, 2007), and (iv) the frequency of word use in daily discourse (Kittredge et al., 2008). Inspired by these results in human cognition, we seek to investigate the impact of personal names from different parts of the world on the linear spatial reasoning ability of large language models - a task that ought to be neutral to the choice of the names themselves.

The rise of the large language models promises to revolutionize differentiable approaches to natural language processing. Models such as BERT (Devlin et al., 2018), T5 (Raffel et al., 2020), GPT (Brown et al., 2020), OPT (Zhang et al., 2022), PALM (Chowdhery et al., 2022) and BLOOM (BigScience, 2022) yield results close to the state-of-the-art on popular benchmarks in

2022 Trustworthy and Socially Responsible Machine Learning (TSRML 2022) co-located with NeurIPS 2022.

several language tasks. However, such models are susceptible to adversarial attacks (Wang et al.) and reduced accuracy on out-of-distribution data (Du et al., 2021). Bias and toxicity evaluations of such models are now standard with a variety of benchmarks (Ousidhoum et al., 2021). Inspired by both the neurolinguistic findings about the curious case of personal names in human cognition and the now well-established bias studies in deep learning, we study personal names as a new source of variations in the performance of large language models even when the underlying reasoning task ought to be independent of the choice of personal names.

We create a scalable spatial reasoning task involving $n$ individuals with different names. In this task, we provide the pairwise spatial relationship between entities to the BLOOM family of models, and then ask the model to predict a new relationship among the entities. Using male names popular in the US on the spatial reasoning task with four individuals, the accuracy of the BLOOM model grows from 0.3 to an impressive 0.83 as the number of parameters increases from 560 million to 146 billion. A random guess would achieve an accuracy of only 0.33 on the task. Thus, an increase in the model size leads to an improvement in the accuracy of the model's accuracy in reasoning about the spatial relationships, and sufficiently large models exhibit very good spatial reasoning capability.

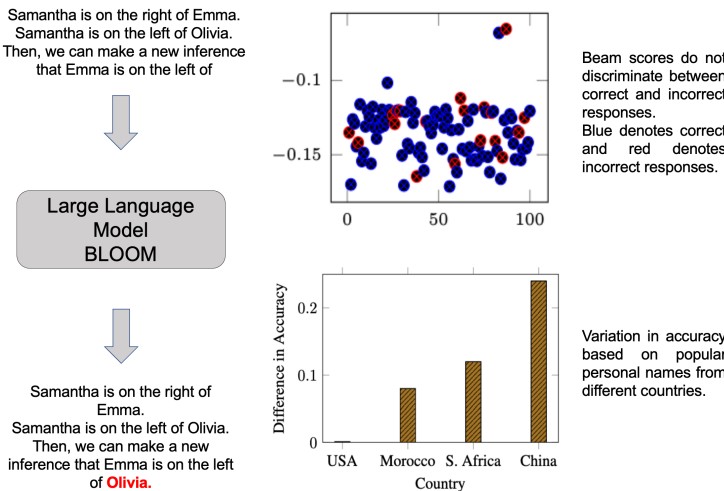

Figure 1: Overview of our approach. (left) BLOOM model for solving the spatial reasoning task. (top right) Beam scores from the largest BLOOM model for correct (blue) and incorrect (red) responses. (bottom right) Variation in response from the largest BLOOM model on personal names from different countries.

**Variation of accuracy based on names:** The accuracy of the BLOOM models on this spatial reasoning task varies depending on the source countries of the proper names even though the reasoning task is completely invariant to the choice of the names. The difference in accuracies for popular proper names from USA, Morocco, South Africa and China are shown in Fig. 1 (bottom right). When we change the names in the reasoning task using the names common in US, the variance is 0 but it increases to 0.08 for names from Morocco, 0.12 for South Africa, and 0.24 for China. We conjecture that the difference in accuracy may be related to the use of unicode characters while writing names from international countries and perhaps the fewer occurrence of such names in the training data. Further, the difference in accuracies depending on the choice of proper names is generally exacerbated in larger BLOOM models.

**Poor calibration of BLOOM models on spatial reasoning**: We observe that the conditional log probabilities or beam scores of BLOOM predictions are not well-calibrated, and the scores do not discriminate between correct and wrong responses. The beam scores of 100 responses from BLOOM-176B are shown in Fig. 1 (top right). The scores for the correct (incorrect) responses are shown in blue (red). As shown in the figure, the distributions of the two scores can not be separated. Thus, this approach for uncertainty quantification is not sufficient to detect when the model makes wrong predictions due to name bias in this setting.

We believe that our results show that there is a need to develop training strategies and inference algorithms that can mitigate emergent bias for large language models in such settings, where performance in the underlying task ought to be clearly independent of the source of the bias.

## 2 Spatial Reasoning in BLOOM Large Language Models

### 2.1 Spatial Reasoning Tasks

Spatial reasoning has been a topic of sustained interest in representational learning. Qualitative reasoning using spatial cardinal directional representations has been investigated for at least three decades (Frank, 1991; Teresa Escrig & Toledo, 1998). Spatial reasoning for textual data (Weston et al., 2015) using deep learning models (Le et al., 2020) has made such rapid progress that has led to the need for creating new and more challenging benchmarks (Shi et al.).

---

**Algorithm 1:** Synthesis of Spatial Problems

---

**Input:** List of popular names $p_1, p_2, \ldots p_n$, Set of left-of relations $S_l = \{(p_i, p_j)|p_i$ is on left of $p_j\}$,
    Set of right-of relations $S_r = \{(p_i, p_j)|p_i$ is on right of $p_j\}$, Query Pair $(p_k, p_l)$
**Output:** Correct Response $R$

1   $V \leftarrow \{p_1, p_2, \ldots p_n\}, E_l \leftarrow \phi, \; E_r \leftarrow \phi$            // Graph
2   **for** $(p_i, p_j) \in S_l$            // Add left-of relations
3   **do**
4     $E_l \leftarrow E_l \cup (p_i, p_j), E_r \leftarrow E_r \cup (p_j, p_i)$

5   **for** $(p_i, p_j) \in S_r$            // Add right-of relations
6   **do**
7     $E_r \leftarrow E_r \cup (p_i, p_j), E_l \leftarrow E_l \cup (p_j, p_i)$

8   $E_l^0 \leftarrow E_l, E_r^0 \leftarrow E_r, i = 0$ **while** $i = 0$ *or* $E_l^i \neq E_l^{i-1}$      // transitive closure
9   **do**
10    $E_l^{i+1} = E_l^i$ **if** $(p_i, p_j) \in E_l^i$ *and* $(p_j, p_k) \in E_l^i$ **then**
11      $E_l^{i+1} \leftarrow E_l^{i+1} \cup (p_i, p_k)$
12    $i \leftarrow i + 1$

13   $i = 0$ **while** $i = 0$ *or* $E_r^i \neq E_r^{i-1}$            // transitive closure
14   **do**
15    $E_r^{i+1} = E_r^i$ **if** $(p_i, p_j) \in E_r^i$ *and* $(p_j, p_k) \in E_r^i$ **then**
16      $E_r^{i+1} = E_r^{i+1} \cup (p_i, p_k)$
17    $i \leftarrow i + 1$

18   **if** $(p_k, p_l) \in E_l^i$ **then**
19    $R \leftarrow p_k$ is on the left of $p_l$

20   **if** $(p_k, p_l) \in E_r^i$ **then**
21    $R \leftarrow p_k$ is on the right of $p_l$

22   **if** $(p_k, p_l) \notin E_l^i$ *and* $(p_k, p_l) \notin E_r^i$ **then**
23    $R \leftarrow$ Relationship between $p_k$ and $p_l$ not known.

---

We create a simple spatial reasoning task where $n$ individuals are located on a straight line. The task specifies the relative location, left or right, of one person with respect to another. Finally, the task requires that we make a new hitherto undeclared inference about the relative position of individuals. We keep the reasoning problem simple because our goal is not to evaluate the limits of the reasoning capability of these large language models, but instead to analyze their bias with respect to the used proper names in defining the reasoning problem.

The challenge problems in our tasks are created algorithmically. See Algorithm 1 on the right. Given $n$ individuals and a query pair $p_k, p_l$, we construct a left-graph $G_l = (V, E_l)$ of $n$ nodes. For every pair of nodes $p_i, p_j \in V$, we add the directed edge $(p_i, p_j)$ to the left-graph if $p_i$ is known to be on the left of $p_j$ according to the stated facts in the generated task. Finally, we compute the transitive closure $G_l^* = (V, E_l^*)$ of the left-graph.

If $(p_k, p_l) \in E_l^*$, the given facts in the generated task are adequate to show that $p_k$ is to left of $p_l$, we keep this problem in our task and note that the correct response states that $p_k$ is to left of $p_l$. Similarly, we create a right-graph where the edges denote that an individual is to the right of another, compute its transitive closure and determine whether the stated facts are enough to conclude that $p_k$ is to the right of $p_l$. An illustrative automatically generated task is shown below with the response from the BLOOM-176B model colored in red.

> John is on the right of David. James is on the left of David. James is on the left of John. Joseph is on the right of James. John is on the left of Joseph. Then, we can make a new inference that David is on the left of Joseph.

## 2.2 BLOOM Large Language Models

The recent increase in the size of large language models using distributed GPUs has led to impressive performance on diverse tasks, such as finishing sentences (Zellers et al., 2019), commonsense story cloze (Mostafazadeh et al., 2016), physical commonsense reasoning (Bisk et al., 2020), challenging question answering problems (Clark et al., 2018), open book question answering (Mihaylov et al., 2018), Turing tests based on correct word disambiguation (Levesque et al., 2012; Sakaguchi et al., 2021), and more challenging general-purpose language understanding (Wang et al.).

A variety of large language models have been trained on internet-scale corpora of text in multiple languages and code bases. These models include BERT, T5 , GPT, OPT, PALM and BLOOM. A few of these models, such as GPT3, are only accessible via an API while all the trained weights for other models, such as BLOOM, are available to the public at large.

Our experimental studies have focused on BLOOM as it is an open-science open-access model that has been trained using data in multiple languages and is the result of an international effort. BLOOM has been trained on the Jean Zay Public Supercomputer provided by the French government and is readily available to the public (BigScience, 2022). BLOOM is a family of large-language models ranging from 560 million parameters to 176 billion parameters, and provides an effective platform for evaluating the variations in our spatial reasoning task. BLOOM models have been trained on substantial text from multiple languages.

## 2.3 Efficacy of BLOOM and Variation due to Choice of Names

We observe that BLOOM is capable of solving the spatial reasoning task with zero-shot prompting in many cases. An example can be seen in the illustration in Fig. 1. We consider reasoning tasks with different complexities by changing the number of individuals over which spatial reasoning needs to be performed. Each language task was performed 100 times to compute the average accuracy of the task and a fixed number of token predictions were sought from the model. Our list of popular names in different countries was first obtained from Wiktionary and then supplemented by the popular name lists available on Wikipedia. We used the English language transliteration of the names instead of using the names in their original script. As an example, names from China were written in English but accents were included, thereby exposing the larger set of possible phonologies (James & Fogler, 2007). We performed experiments on the BLOOM family of large language models on a server with 256 AMD cores, 2 TB of RAM and 8 Nvidia A100 80GB GPUs.

**Spatial reasoning over 3 individuals.** Figure 2 shows the performance of the BLOOM family of models on our spatial reasoning task for 3 male names drawn from the popular names of different countries. The accuracy of the BLOOM family rsies in a sustained manner from a maximum of $0.52$ for the 560 million parameter model to a maximum of $1.0$ for the 176 billion parameter model.

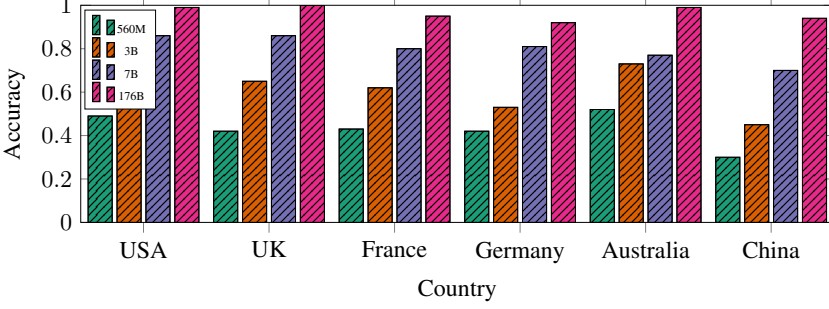

Figure 2: Reasoning with 3 male names using 4 different BLOOM models with increasing model parameters.

**Spatial reasoning over 4 individuals.** Figure 3 shows the performance of BLOOM family of models on the spatial task with 4 male names drawn from the popular names of different countries. The 560 million parameter model has a maximum accuracy of only $0.3$, which is below random chance. However, the 176 billion parameter model has an accuracy of $0.83$ for the United States.

**Accuracy Variation due to Choice of Names.** While the significant variation of the model accuracy on the choice of the names on an unrelated spatial reasoning task is itself a source of

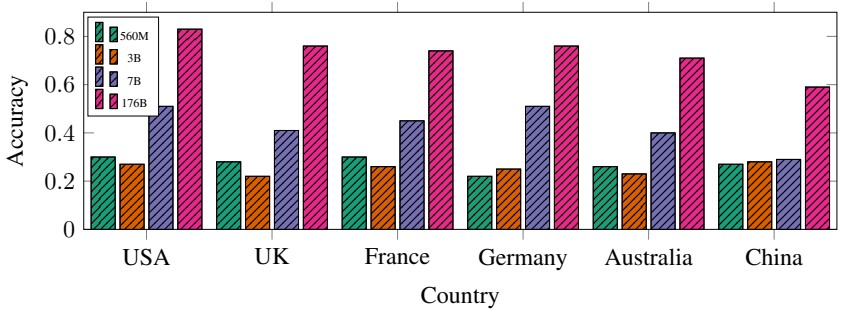

Figure 3: Efficacy of spatial reasoning with 4 male names popular in different areas using BLOOM models.

concern, we observe that the variation is emergent and it generally increases in magnitude as the models grow in size. We quantitatively evaluate the variation as the difference between the accuracy for a given geographical area and the maximum accuracy for a model of that size. For example, for 4 individuals, names drawn from China have an accuracy of 0.59 while names drawn from USA have an accuracy of 0.83 for the BLOOM-176B model. Hence, the variation for China is the difference of the two accuracies, i.e., 0.24. This high variation indicates relatively low accuracy for the corresponding choice of names even when the underlying reasoning task is the same and should not depend on the names being used.

We analyze the variation on a challenging spatial reasoning task with 5 individuals. Figure 4 shows that the BLOOM-176B model shows a variation of 0.03 for the United States. However, the variation for China, India, and South Africa becomes as high as 0.14, 0.17, 0.10. We also notice a high variance for Canada and a low variance for Morocco, which suggests that other confounding factors not being analyzed by us may also be present in our observations.

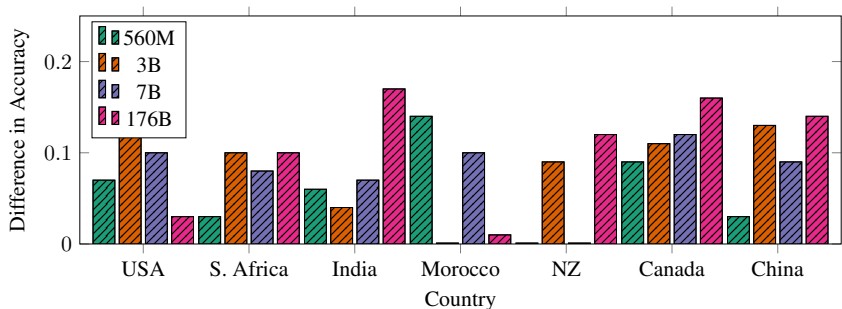

Figure 4: Difference between the best accuracy for a given model and the accuracy for popular male names drawn from a geographical region. This spatial reasoning task uses 5 individuals.

These variations may be explained using various reasons: (i) different term-frequency in training data, (ii) lack of descriptive value of a personal name that can be learned using a large text corpora, and (iii) the potentially larger set of possible phonologies in different languages, especially with our use of accents and other unicode English-like characters while transliterating from international names. However, none of this would be a problem for classical graph-based analysis algorithms. As classical algorithms are replaced by increasingly complex and often proprietary and closed large language models, the variation from proper names needs to be addressed to avoid biased analyses.

## 3 Uncalibrated Model Scores for Predictions

Since the BLOOM models produce significantly different accuracies on the spatial reasoning task for different choice of names, it becomes even more important to understand if the model is predicting a correct response on a given query. A natural approach to quantify uncertainty of a model is to use the log probability of the response predicted by the model and threshold it to decide if a response is reliable or not. We show that the BLOOM models are not well-calibrated and produce similar conditional log probabilities for both correct and incorrect responses in this context.

The BLOOM models, like other large language models, compute the beam score or the conditional log probability of the response being predicted. In a well-calibrated model, a correct output will be associated with a high probability and an incorrect output will correspond to a relatively lower probability. This will allow the model to be deployed in a practical setting.

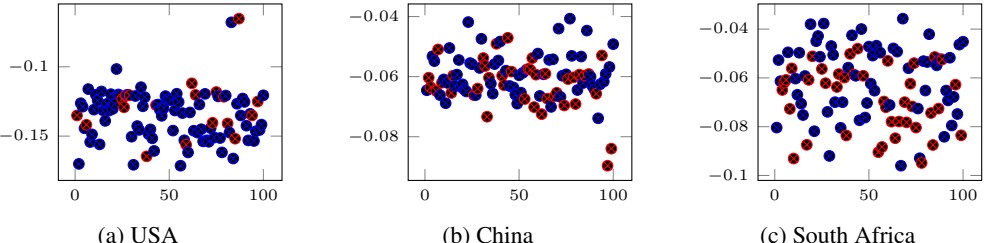

(a) USA        (b) China        (c) South Africa

Figure 5: The beam score or conditional log probabilities for the top response from the model using male names. The X axis represents the index of the query and the Y axis represent the score. Red color represents an incorrect response, while a blue color represents a correct response.

Fig 5 shows the beam score or log probability corresponding to a typical response from the BLOOM-176B model on male names from the US, China and South Africa. The horizontal axis represents the index of the query and the vertical axis represents the beam score or the log probability of the response.

## 4  Conclusions and Future Work

We make three contributions in this paper. First, we create a simple spatial reasoning task where $n$ individuals are located on a straight line. The task specifies the relative location, left or right, of one person with respect to another. Finally, the task requires that we make a new hitherto undeclared inference about the relative position of individuals. We then observe that BLOOM and other large language models have the emergent capability of solving this linear spatial reasoning task as these models become larger. Second, we make empirical observations on the choice of names and their influence of model accuracy in our linear spatial reasoning task. While there has been significant work on evaluating explicit bias in the predictions from large language models, we make a new observation that BLOOM large language models are substantially influenced by the choice of personal names on the linear spatial reasoning task. Third, we observe that the beam score or conditional log probabilities of predicted tokens have poor correlation with the correctness of the model response, and cannot be relied upon as a metric for uncertainty quantification on this task.

The analysis and observations in this paper open up a couple of exciting directions for future research. First, it will be interesting to compare the performance of BLOOM with other large language models that were not designed by an international team of researchers. Since the trained weights of many of the state-of-the-art models are not available to the public at large, such proprietary models need similar internal auditing. Second, we have employed English transliterations with accents and diacritics that may explain the increased phonology and reduced accuracy for international names. Experimental investigations that avoid accents and diacritics may be needed to conclude that different choice of names lead to different accuracies even when the underlying search space is unaccented English. Third, our work has focused on personal names, and it may be interesting to pursue a wider study using different proper nouns, such as places and brand names. It may also be interesting to replace proper names with hitherto unknown sequences of characters as a test for proper names of different complexity.

## Acknowledgement

The authors acknowledge support from DARPA cooperative agreement #HR00112020002. The views, opinions and/or findings expressed are those of the authors and should not be interpreted as representing the official views or policies of the Department of Defense or the U.S. Government.

## 5  Ethics Statement

Our approach brings to light a potential concern with BLOOM and possibly other large language models where their accuracy is potentially affected by the choice of personal names, even on tasks that are completely independent of the choice of names.

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
