# OpenReview forum: "Responsible Reasoning with Large Language Models and The Impact of Proper Nouns"
_NeurIPS.cc/2022/Workshop/TSRML — TSRML2022_

### Official Review · Reviewer_QpMX · 2022-10-17
**Interesting work on spatial reasoning for LLMs, missing some important details**

**Overall Rating:** 5

**Summary:**

This paper introduces a measure of the difference in accuracy when different names corresponding to different demographic groups are substituted in for a spatial reasoning task. They find that names from different countries lead to large differences in accuracy on this task.

**Strengths:**

- The direction on spatial reasoning is interesting and it's useful to know these models exhibit accuracy disparities across names associated with different demographic groups

**Weaknesses:**

- More description on experimental procedure is needed. What are the names? For chinese names, do you use pinyin or characters? If the latter, does the bloom tokenizer support encoding chinese characters or does it only support latin-1 encoding for instance, explaining the large difference in accuracy?
- Additional experimental evaluation the limitation would be useful as well. If you introduce few shot examples, do these problems go away?
- The results on calibration are somewhat confusing. If these models are not even correct in the first place, how could we expect them to be well-calibrated?

**Overall Recommendation:**

This is an Interesting direction, but some more work could be needed here to complete the story.

**Review Confidence:**

4: The reviewer is confident but not absolutely certain that the evaluation is correct

---

### Official Review · Reviewer_mTAF · 2022-10-19
**Well-scoped investigation into previously-unstudied bias of language models**

**Overall Rating:** 8

**Summary:**

This paper analyzes how the use of different proper nouns in spatial reasoning tasks affects language models' performance on those tasks. It finds that models perform best on spatial reasoning tasks that contain popular American names and worst on tasks with names from other cultural context (e.g. Chinese).

**Strengths:**

- The authors present a well-scoped analysis of a specific problem: name-based bias in spatial reasoning tasks.

- Results are clearly and systematically presented throughout the paper.

- The result in Section 4 seems significant. I understand that the paper is short, but the authors should elaborate on this result, either in the appendix or a future, non-workshop submission. Is the implication of this that conditional log-probability scores are meaningless for measuring BLOOM's correctness on spatial reasoning tasks? Does it apply more broadly? If the latter, this seems like a problem.

- It would be interesting to study if this phenomenon generalizes to other types of tasks in BLOOM. Have the authors considered this? Only consideration of other proper nouns, not other tasks, is mentioned in the conclusion.

**Weaknesses:**

- "A socially responsible large language model would ...": This sweeping moral claim (both in line 4 and in title) is unnecessary. The authors show that there is an interesting link between models' spatial reasoning and the names of people involved. This is both an interesting and concerning result - and, I think, most people would agree that it holds moral/societal implications that need to be addressed. However, without any provided definition of "socially responsible," it is difficult to reason about this broader moral judgement about how models should behave. Either the authors should provide a definition for this term or omit this sort of language.

- It seems the authors could have dug deeper into the training data of BLOOM to assess underlying reasons for the observed name/spatial reasoning behavior. For example, in line 132, the authors note that BLOOM models have probably been trained on data from multiple countries. Can you provide a definitive answer for this question? Furthermore, it would be interesting to estimate how many names a model sees during training. If it's not very many, then the observed behavior might make more sense.  In exploring the [Notion page](https://bigscience.notion.site/BigScience-214dc9a8c1434d7bbcddb391c383922a) for BLOOM's development, I found this link to the [BigScience data catalogue](http://23.251.145.180:8501/) (which was supposedly used to train BLOOM). Might be helpful.

- Nitpicks
    - Algorithm 1 is never mentioned in the text.
    - Legend covers 3B USA score in Figs 3 and 6.
    - Why are there two elements in the legends of Figures 3-6 for each model size? Do they mean anything? If not, please remove. They're distracting.


**Overall Recommendation:**

I recommend this paper be accepted. If and when the authors choose to expand on this paper and submit it to a non-workshop venue, I recommend they take the weaknesses I highlighted into consideration.

**Review Confidence:**

4: The reviewer is confident but not absolutely certain that the evaluation is correct

---

### Decision · Program_Chairs · 2022-10-23

Accept